# Insights into Transcriptomic Differences in Ovaries between Lambs and Adult Sheep after Superovulation Treatment

**DOI:** 10.3390/ani13040665

**Published:** 2023-02-14

**Authors:** Qingwei Wang, Xiaofei Guo, Dawei Yao, Biao Wang, Yupeng Li, Jinlong Zhang, Xiaosheng Zhang

**Affiliations:** 1Institute of Animal Science and Veterinary, Tianjin Academy of Agricultural Sciences, Tianjin 300381, China; 2Tianjin Key Laboratory of Animal Molecular Breeding and Biotechnology, Tianjin 300381, China; 3Tianjin Engineering Research Center of Animal Healthy Farming, Tianjin 300381, China

**Keywords:** LncRNAs, mRNAs, juvenile superovulation, sheep

## Abstract

**Simple Summary:**

By treating one-month old and adult Hu sheep with superovulation, we found that one-month-old lambs are able to produce more follicles than adult sheep. Nevertheless, our previous study revealed that oocytes derived from juvenile animals tend to be of poor quality, which means that they are less likely to mature and develop into embryos. By treating sheep with exogenous ovulation-inducing hormones, we compared the number of ovulation points and the serum hormone levels in both the lamb and adult groups. In order to reveal the mechanism of abnormal ovulation numbers and oocyte quality induced by hormone treatment in the lamb group, a high-throughput sequencing technique was used to identify the differences in long non-coding RNA (lncRNA) and messenger RNA (mRNA) expression in the ovaries in the lamb and adult groups. The conjoint analysis of lncRNA and mRNA revealed that OaPDGFR, XR_003588840.1, and its target gene OaLRP1, may be the key candidate genes in promoting the number of ovulations in lamb groups.

**Abstract:**

Superovulation technology shows a great potential for shortening breeding time. Using the juvenile superovulation technology, juvenile animals can generate more follicles than adult animals. By sequencing using high-throughput methods, we studied and described differentially expressed (DE) long non-coding RNA (lncRNAs) and messenger RNAs (mRNAs) in the ovaries of young and adult sheep. Herein, 242 DE lncRNAs and 3150 DE mRNAs were screened. Through GO and KEGG analyses, we obtained genes related to ovarian/follicle development and ovulation in DE mRNAs, including *OaFSHR*, *OaLHCGR*, *OaLDLR*, *OaZP3*, *OaSCARB1*, and *OaPDGFRA*; through lncRNA-mRNA correlation analysis, we found that genes associated with ovarian/follicle development or ovulation include: XR_003585520.1, MSTRG.15652.1, XR_003588840.1, and their paired genes *PDGFC*, *LRP5*, and *LRP1*. We observed a synergistic effect between *PDGFR* and *LRP1*. *PDGFR* may play a leading role compared with *LRP1*. The induced *LHCGR* in lambs is higher than in adult sheep, showing more sensitivity to LH. The release of the oocytes was stimulated. Among the three lncRNAs, we found that XR_003588840.1 was significantly different and might perform a regulatory role in ovarian/follicle growth or ovulation.

## 1. Introduction

In breeding domestic animals (sheep, cattle, pigs), the reproductive cycle and generation interval limit genetic advancement. Investigating the reproductive ability of young animals can overcome the previous restricting variables [1]. Since the gonadal system of young animals is immature, primordial germ cells exist in the ovaries. Superovulation must be performed on the donor dam during embryo transfer to obtain many embryos and produce twins in singleton livestock. Almost all technologies relevant to embryo generation and management in ruminants were initially established in sheep and then adapted to other animals [2,3].

The most common use of ovarian stimulation with hormones in sheep is for multiple ovulation and embryo transfer (MOET) [4]. Nevertheless, substantial differences in superovulation responses restrict the implementation of superovulation, particularly in commercial aspects [5,6]. The reproductive condition and record of females, along with the season and photoperiod/melatonin release, significantly impact the superovulation results [7,8]. Under tropical and subtropical conditions, sheep receiving ovarian stimulation are less susceptible to photoperiodic variations than those living under temperate conditions [9]. However, the association has yet to be entirely demonstrated; undernutrition can also affect the embryo by impairing follicle/oocyte competency [10], luteal function [11], and early embryogenesis [12,13,14]. Variations in superovulation responsiveness are restricting elements in ovine MOET projects.

Follicular wave patterns (specifically follicular domination) influence ovarian responsiveness to gonadotrophin super stimulation and, consequently, the efficacy of MOET strategies. When stimulated by superovulation hormones, prepubescent females of the appropriate age can generate significantly more antral follicles compared to adult females [15]. Nonetheless, these antral follicles in prepubescent females fail to ovulate naturally, without a clear cause, necessitating follicular aspiration on the juvenile ovary to recover the oocytes [9]. After maturation and fertilization in vitro, the resulting embryos were implanted into the recipients. This method, juvenile in vitro embryo transfer (JIVET), is complex compared to MOET, used in adult females, and embryos generated from juvenile oocytes have lower development probabilities [2]. The initial count of small follicles at FSH therapy correlates with the ovulation rate in superovulated ewes. When progestogens are used, some effect variations can be expected in the large follicles. In addition, the use of progestins can alter the growth pattern of follicles and increase the atresia of large follicles [16].

During the past few years, a significant amount of research has been conducted on sheep breeding [17], but the mechanism for producing different physiological states in sheep ovaries after superovulation and estrus synchronization treatment has not been fully understood [18]. The development of oocytes and embryos from juvenile animals remains at a reduced level compared to that of adults, although more follicles can be acquired in juvenile animals. Therefore, we hypothesized that the ovarian growth rate in juvenile animals might be associated with the expression levels of lncRNA and mRNA, limiting its large-scale applicability [19]. Herein, an exogenous hormonal therapy was used to induce Hu sheep to contrast the variations in the impacts of ovarian superovulation and serum hormonal levels in young and adult sheep. Therefore, differentially expressed (DE) lncRNAs and mRNAs from the ovaries of young and adult Hu sheep were examined utilizing high-throughput sequencing technology to detect the growth pathway of numerous follicles and inferior oocyte quality in juvenile ovulation. The data will identify the eligible lncRNAs and mRNAs to anticipate ovarian superovulation capacity and oocyte quality.

## 2. Materials and Methods

### 2.1. Research Ethics

All sheep-related experimental protocols were authorized from the Science Research Department of the Tianjin Academy of Agricultural Sciences (TAAS; Tianjin, China). This study was granted ethical approval from the Animal Welfare Committee of TAAS, with access No. 2020009.

### 2.2. Experimental Design, Superovulation Techniques, and Sample Collection

The experimental grouping and superovulation treatment scheme used in this paper were performed as described in our previously published studies [19]. We used the Chinese Hu sheep, treated with superovulation, and two groups were defined: the lamb group (1 month old, n = 6) and the adult group (24 months old, n = 6), and serum and ovaries were collected from each group. All serum was frozen at −80℃. A serum product was utilized to determine the levels of FSH, luteinizing hormone (LH), progesterone (P_4_), and estradiol (E_2_). Each hormone was tested by employing an iodine [^125^I] radioimmunoassay kit (BNIBT, Beijing, China). Then, three sheep in each group were randomly picked for ovary sample collection. After anesthesia, a midline laparotomy was performed in the supine position to expose the left ovary. The ovaries are quickly recovered and snap frozen in liquid nitrogen.

### 2.3. RNA Extraction, Library Construction, and Sequencing

For ovaries frozen in liquid nitrogen, each was smashed into powder, and subsequently recovered for RNA extraction utilizing the Trizol technique. After extracting the total RNA from a sample, ribosomal RNA was eliminated to optimize the maintenance of all coding and noncoding RNA [20]. Therefore, the fragmented RNA was utilized as a template to generate the first strand of cDNA with random hexamers, to which was then added buffer, dNTPs (dUTP instead of dTTP), and RNase H. The second strand of cDNA was synthesized with DNA polymerase I, purified using a QiaQuick PCR kit, isolated with EB buffer, end-repaired, applied with base A, and then degraded using a UNG (Uracil-N-Glycosylase) enzyme chain. The fragment size was determined using agarose gel electrophoresis, and PCR amplification was carried out. Illumina HiSeqTM 4000 was used to sequence the completed sequencing library.

### 2.4. Sequencing Data Analysis

To ensure the data quality, the raw data were filtered before additional analysis. FASTQ (version 0.11.5) was employed to quality control (QC) the raw reads, removing the adapter-containing reads, reads with an N ratio greater than 10%, and low-quality reads. Therefore, TopHat2 (version 2.1.0.) software was used to map the filtered raw reads (clean reads) with the sheep reference (https://www.ncbi.nlm.nih.gov/datasets/genomes/?taxon=9940&utm_source=gquery&utm_medium=referral (accessed on 20 September 2022)).

### 2.5. Identification of Candidate lncRNAs and mRNAs

The Cufflinks program was employed to compare the total number of reads to the genome for the alignment, annotation, classification, and screening of transcripts and genes. LncRNAs were identified according to the next criteria as length ≥ 200 nt, count of exons ≥ 2, FPKM (fragments per kilobase of exon per million mapped reads) ≥ 0.1, and using four kinds of protein-coding potential analysis. The software CNCI (version 2.0), CPC (version 2.0), PFAM (version 5.2.1.1), and CPAT (version 1.2.2) removed transcripts with coding ability to obtain lncRNAs information in the ovarian samples [21]. StringTie was employed to reconstruct transcripts and calculate the expression levels of all mRNAs in each sample [22]. Furthermore, the sequencing depth, and then the length of the gene or transcript, were corrected. After obtaining the FPKM (version 5.2.1.1) value of the genes, subsequent analysis was performed.

Differential expression (DE) analysis of ovarian lncRNA and mRNA between the adult and lamb groups was performed by DESeq (version 1.18.0) software. In the process of screening DE lncRNAs, the |Log_2_ (Fold Change)| ≥ 2 times and the false discovery rate (FDR) < 0.05 were used as standards. For DE mRNAs, the |Log_2_ (Fold Change)| ≥ 1 time and the FDR < 0.05 were used as standards.

### 2.6. Functional Analysis

Separate Gene Ontology (GO) and Kyoto Encyclopedia of Genes and Genomes (KEGG) enrichment analyses were conducted using the anticipated target mRNAs of the DEGs and DE mRNAs. These DE mRNAs were blasted against the GO and KEGG databases of all recognized animals. The gene counts for the GO and KEGG terms were measured, and a hyper-geometric test was performed to identify GO terms and KEGG pathways that were significantly enriched. Significant enrichment was defined as an adjusted *p*-value (Q-value) < 0.05.

### 2.7. Real-Time qPCR Validation

The PrimeScriptTM RT reagent kit (TaKaRa, Dalian, China) was used to conduct reverse transcription for the qPCR test of mRNAs and lncRNAs. Moreover, qPCR was carried out utilizing the SYBR Green qPCR Mix (TaKaRa, Dalian, China) and a RocheLight Cycler R 480 II system (Roche Applied Science, Mannheim, Germany) as follows: initial denaturation at 95 °C for 5 min, 40 rounds of denaturation at 95 °C for 5 s, and annealing at 60 °C for 30 s. qPCR and RNA-seq were mapped utilizing Prism (version 8.3.0).

### 2.8. Statistical Analysis

In order to verify the reliability of the transcriptome data, we analyzed the correlation between RNA-seq data and qRT-PCR data. In this study, each sample was tested three times. The qRT-PCR data used ΔΔCt (ΔCt reference-ΔCt target) and 2^-ΔΔCt^ formulas to obtain standardized gene expression. The relevant fitting curve data between qRT-PCR and RNA-seq gene expression results are found in Appendix A. Finally, SPSS (v 20.0) was used to statistically analyze the experimental data.

## 3. Result

### 3.1. Sequence Data Summary

Three biological replicates of each group (lamb and adult) were chosen to create lncRNA and mRNA libraries for sequencing investigation. A total of 468,829,966 bases were obtained from the 6 sequence libraries. After filtering, a total of 467,988,964 clean labels were generated (Table 1). On average, 97.11% of the clean reads were mapped on the sheep genome, with an average unique mapped rate of 93.88%. The average Q20 and Q30 value was 97.72%, and 93.48%, respectively. The GC content was 44.40% (Table 1). Additionally, the Pearson’s Correlation Analysis (PCA) was performed based on the identified mRNA and lncRNAs. The results showed that all the biological repeats exhibited a good correlation, and samples in the same group could be gathered together (Appendix A). In total, 3150 DEmRNAs were identified, including 1359 and 1791 up- and downregulated DEmRNAs, respectively (Figure 1A). In total, 242 DElncRNAs were identified, including 108 and 134 up- and downregulated DElncRNAs, respectively (Figure 1B).

### 3.2. Functional Analysis

GO and KEGG analyses for DEmRNAs were conducted. The results showed that 393, 675, and 3952 GO terms were significantly enriched for the CC, MF, and BP ontologies, respectively. Figure 2A shows the top 20 enriched GO terms. GO terms related to the ovulation cycle process and the ovulation cycle (GO:0022602), and GO terms related to follicular development (GO:0042698) were enriched in the BP ontologies. *OaZP3* and *OaPDGFRA* were involved in the two GO terms and were highly and lowly expressed in lambs, respectively (Figure 2B). Subsequently, the KEGG enrichment analysis pathway was evaluated according to the DE mRNAs. A sum of 333 KEGG pathways were enriched, of which 48 pathways were significantly enriched (Q-value < 0.05). Ovarian steroidogenesis (ko04913) was the most significantly enriched of all the enriched KEGG pathways. Ovarian steroidogenesis is one of the critical pathways affecting follicular development, so the DE mRNA in the ovarian steroidogenesis pathway is particularly critical. In total, 19 DE mRNAs were identified in the ovarian steroidogenesis pathway, including *OaCPY19*, *OaCYP17*, *OaLDLR*, *OaLHCGR*, and *OaSCARB1* (Figure 2C). Of all the DEmRNAs, *OaFSHR*, *OaLDLR*, *OaLHCGR*, and *OaSCARB1* were the essential membrane proteins (Figure 2C).

### 3.3. LncRNA-mRNA Interaction Analysis

The expression patterns of lncRNAs exhibited a significant effect on nearby (cis) or distal (trans) protein-coding genes. LncRNA-mRNA interactions were beneficial for elucidating the biological activities of DE lncRNAs. We studied the possible interactions between lncRNA and mRNA transcripts to identify the lncRNAs and their potential roles. The study of cis-/antisense-acting targets was conducted by correlating the expression profiles of DE-lncRNAs and DE-mRNAs. In total, 561 and 2625 lncRNA-mRNA pairings engaged in antisense- and cis-regulation, respectively, were found. Accordingly, 5 antisense-acting lncRNA-mRNA interactions involving 5 DE lncRNAs and 4 DE mRNAs; 30 cis-acting lncRNA-mRNA interactions involving 30 DE lncRNAs and 27 DE mRNAs were identified (Table 2).

No ovarian and follicular development-related DEmRNAs were identified; however, three mRNAs, *OaLRP1*, *OaPDGFC*, and *OaLRP5*, showed interactions with XR_003588840.1, XR_003585520.1, and MSTRG.15652.1, respectively. XR_003588840.1 showed significantly higher expression in adults than that in lambs (Figure 3). Moreover, *OaLRP1*, *OaPDGFC*, and *OaLRP5* were unrecognized as DE genes because their |Log_2_(Fold Change)| < 1.0 but >0.5, indicating its potential functionality.

### 3.4. Quantitative Real-Time PCR (qRT-PCR) Confirmation

The RNA expression level was determined using the ∆∆−2CT method. qRT-PCR was repeated three times for each sample. GAPDH was used as a reference gene. The expression levels of the LncRNAs and their target genes are depicted in Figure 2. The qRT-PCR results were used to validate the RNA-seq results. The RNA-seq and qRT-PCR correlation analysis results showed a very good match between the RNAseq and qRT-PCR methods. The R2 > 0.90, indicated a strong correlation.

## 4. Discussion

Compared with adult sheep, gonadotropin superovulation in lambs can stimulate follicle growth and obtain more oocytes for in vitro embryo production [23], which has a great application in animal husbandry. However, in vitro embryonic development potential derived from lamb oocytes is significantly reduced [24,25]. The causes of oocyte dysplasia in juvenile animals are not fully understood and remain to be studied. In this study, we identified lncRNA sets and transcriptome data from the ovaries of overostracized lambs and adult sheep to explore the mechanism of lncRNA development in lamb oocytes. We screened 242 DE lncRNAs and 3150 DE mRNAs in the ovaries of lambs and adult sheep. Through GO and KEGG analyses, we correlated these DE mRNAs related to ovarian/follicle development and ovulation, including OaFSHR, OaLHCGR, OaLDLR, OaZP3, OaSCARB1, and OaPDGFRA, while through lncRNA-mRNA correlation analysis, we found XR_003585520.1, MSTRG.15652.1, XR_003588840.1, and their paired genes; PDGFC, LRP5, LRP1 were associated with ovarian/follicle development and ovulation. We also observed a synergistic effect between PDGFR and LRP1. Among the three lncRNAs, we found that XR_003588840 was significantly different and might perform a regulatory role in ovarian/follicle growth or ovulation.

Under the stimulation of exogenous hormones, lambs and adult Hu sheep successfully performed superovulation. The number of follicles and the level of serum reproductive hormones have been described in previous articles. The average number of stimulated follicles in lambs was 70.17 ± 5.14, and in adult sheep, it was 20.17 ± 2.65, revealing a significant difference between them (*p* < 0.01). However, there were no significant differences in the concentrations of FSH, LH, progesterone (P_4_), or estradiol (E_2_) in jugular vein serum of lambs and adult sheep [19]. Furthermore, hormonal stimulation activates biosynthetic processes within the oocyte, which are necessary for enhancing developmental competency [15]. Earl et al. (1998) [26] reported that hormonal inducement of prepubescent donors required smaller hormonal doses compared to adult animals [26] because of lower body weight and better follicular response prior to the emergence of intraovarian processes that control ovulation rate.

In the mRNA sequence data, *LDLR* was identified as an important DE mRNA. It is involved in cholesterol metabolism. Moreover, *LRP1* showed similar functions to *LDLR*, although there was a non-significant difference between the two groups. Cholesterol is a crucial substrate for producing ovarian sex hormones and follicular growth [27]. Most cholesterol in follicular fluid is derived from plasma1. Cholesterol serves as a precursor for tissue-specific steroid hormone production [28]. Steroid-producing tissues have specific cholesterol needs [28].

Consequently, steroidogenic tissues have developed several cholesterol delivery channels, as well as an effective intracellular cholesterol transport mechanism to provide a consistent supply of appropriate cholesterol availability [29]. Across the receptor-mediated endocytosis of lipoprotein particles, the *LDL* receptor regulates cholesterol homeostasis. Enhanced expression of *LRP* delivers cholesterol for steroidogenesis, and it represent a backup system to guarantee that the pathway transfers cholesterol to the steroidogenic tissues [30]. For steroidogenesis, rat ovarian granulosa depends largely on lipoprotein-derived cholesterol, primarily provided by the *LDL* receptor- and scavenger receptor class B type I (SRBI)-mediated pathways. Adult showed had higher *LDLR* gene expression and lower *LRP1* gene expression, and juvenile mice showed lower *LDLR* gene expression and higher LRP gene expression [31]; however, herein, we found that adult sheep had lower *LDLR* gene expression and higher *LRP1* gene expression, and lambs showed LRP1 gene expression and higher *LDL* gene expression. Therefore, in sheep, *LRP*, and *LDLR* ovarian steroids perform a key role in this pathway.

FSH and LH are crucial in controlling ovarian cell development and proliferation [32]. The gonadotropins FSH and LH control the growth and function of the gonads via interacting with their cognate receptors, *FSHR* and *LHCGR*, respectively [33]. For follicles in the terminal follicular phase, FSH induced *LHCGR* expression, and high FSH levels effectively increased *LHCGR* mRNA abundance in pre-sexually mature mice [34,35]. High levels of FSH before sexual maturation activate granulosa cell differentiation insufficiently to allow follicles to reach the telophase of maturation. In infancy, FSH is needed for initial follicle growth to the preantral/early antral phase [36,37,38,39]. Herein, the superovulation process provided exogenous FSH and LH, and the accompanying transcriptome findings revealed that the abundance of *LHCGR* mRNA was significantly higher in young than in adult sheep, perhaps due to the influence of exogenous FSH. Although the supply of exogenous FSH is insufficient to allow the follicle to reach mature telophase, LH supplementation during superovulation activates the ovulatory cascade to stimulate ovulation [40]. Accordingly, we speculated that this theory is applicable in adult sheep and lambs. The induced *LHCGR* in lambs was higher than that in adult sheep and therefore, may be more sensitive to LH, perhaps resulting in more oocytes being stimulated to ovulate in lambs. Laura Torres-Rovira et al. [41] observed the most significant response to ovarian stimulation with exogenous FSH in 50-day-old sheep, with a larger quantity of entirely new follicles in the two groups than those obtained at 195 and 496 days of age, which is consistent with our research. Briefly, the study found that the expression levels of *FSHR* and *LHCGR* in lamb ovaries were significantly elevated compared to those in adult sheep ovaries. The count of mature follicles was increased compared to that in adult sheep.

Inhibition of platelet-derived growth factor (PDGF) increases the count of atretic follicles, and the PDGF system is essential for gonadotropin-induced folliculogenesis. PDGF enhances the in vitro proliferation of theca cells from rat and pig antral follicles, while blocking the LH-induced production of thecal steroid hormones [42]. The characteristics of PDGF and the receptors in stimulating cell proliferation and migration suggested that the signaling of these receptors was crucial for forming the corpus luteum, possibly by influencing the migration and/or proliferation of thecal cells and/or vasculature cells responding to the ovulatory surge of LH [42]. Theca–theca and theca–granulosa cell interactions via PDGF and its receptors promote preantral follicle growth [42]. Inhibition of *PDGFR* in the rat ovary resulted in increased follicular atresia, reduced primary/early follicle and antral follicle formation, and intraovarian vessel size. Our results showed that the gene expression patterns of *PDGFC* and *PDGFRA* in adult sheep were significantly increased compared to those in lambs after superovulation. However, the number of ovulations in lambs is high, but the quality of follicle maturation is inferior to that in adult sheep. This may be because the ovaries and sexual organs of lambs are ill developed. Although they will ovulate under hormone stimulation, the quality of mature follicles will be worse than that in adult sheep.

We also found that *LRP1* is ubiquitously expressed by multiple cells, *LRP1* limits atherosclerosis by blocking the PDGF signaling pathway, and there was PDGF receptor expression in the oocytes and granulosa cells. *LRP1* participated in several physiological activities as a coreceptor and interacted with multiple adaptor proteins via its cytoplasmic domain [43]. Herein, we found a possible relationship between *LRP1* and PDGF.

## 5. Conclusions

This study used RNA-seq data from superovulation treatment lambs and adult Hu sheep to classify LncRNAs and LncRNA target genes. According to sequencing data, OaPDGFR played a leading role when compared with OaLRP1; however, there was a synergistic effect between PDGFR and LRP1. Furthermore, the induced OaLHCGR in lambs was higher than that in adult sheep, so it may be more sensitive to LH and stimulate oocyte maturation. The LncRNAs may play a uniquely role in sheep prolificacy. Finally, we found that XR_003588840.1 was differentially expressed, indicating its potential role in the regulation of ovarian/follicle development. Our results identify a possible candidate LncRNA, as well as reveal the important roles of hormone response and oocyte maturity in addressing barriers for sheep prolificacy.

## Figures and Tables

**Figure 1 animals-13-00665-f001:**
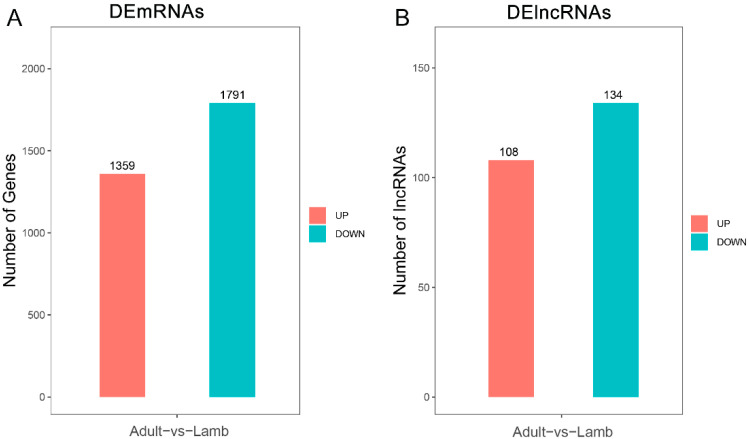
DE lncRNA and mRNA summary. (**A**,**B**) indicate the numbers of DEmRNAs and DElncRNAs, respectively; the red and green bars show the increased and decreased gene expression compared with the lamb group, respectively.

**Figure 2 animals-13-00665-f002:**
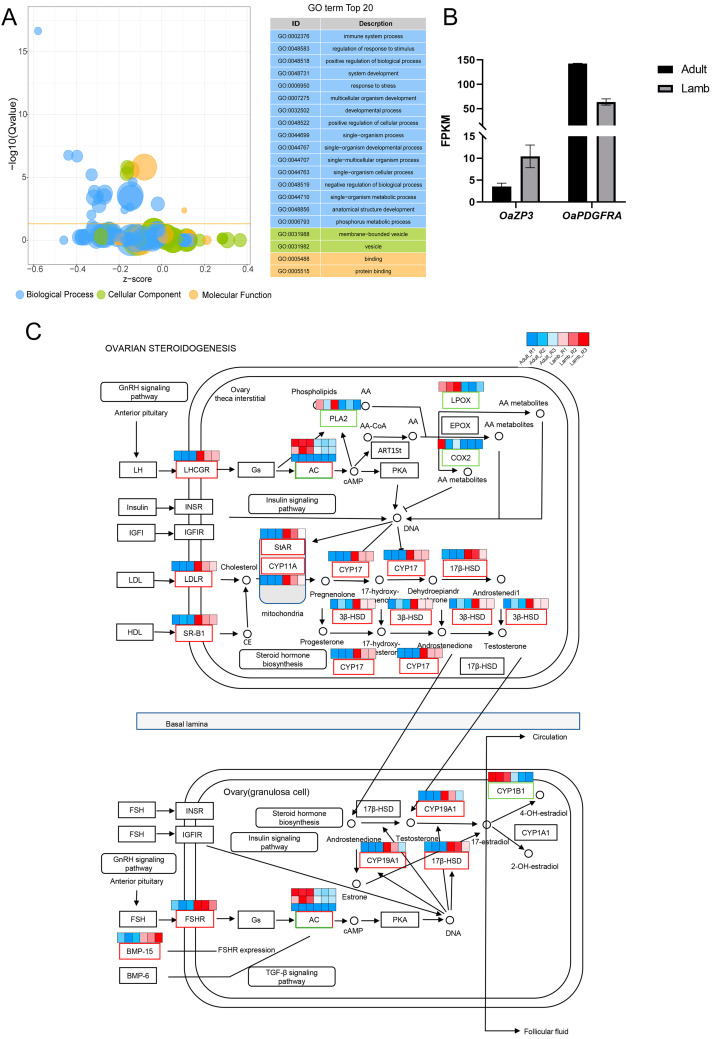
GO and KEGG functional analysis based on DEmRNAs. (**A**) Bubble plot for visualizing the top 20 enriched GO terms; every bubble represents a gene set; the bigger the bubble size, the higher the number of genes in the gene set. (**B**) The expression level of *OaZP3* and *OaPDGFRA*. (**C**) The heatmap of DE mRNAs involved in the ovarian steroidogenesis pathway; the higher the expression level, the redder the color, and the lower the expression level, the bluer the color.

**Figure 3 animals-13-00665-f003:**
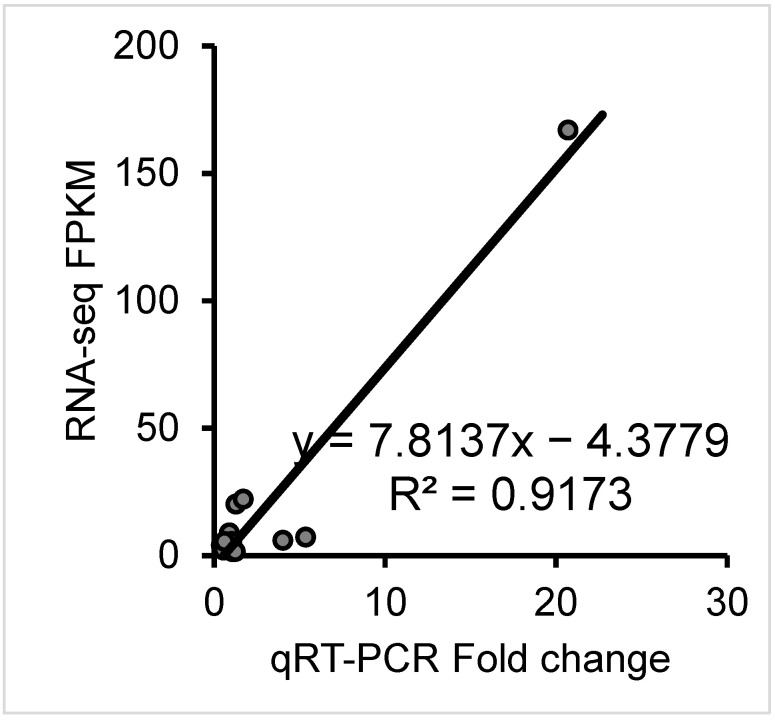
Quantitative real-time PCR (qRT-PCR) confirmation. Correlation analysis between RNA-seq and qRT-PCR methods; Log2(foldchange) values of RNA-seq data (*y*-axis) are plotted against Log2(foldchange) values of qRT-PCR (*x*-axis) data.

**Table 1 animals-13-00665-t001:** Sequence data summary.

Sample	Raw Base (bp)	Clean Base (%)	Total Mapped (%)	Unique Mapped (%)	Q30 (%)	Q20 (%)	GC (%)
Lamb_1	96499362	99.80	96.70	93.46	92.33	97.25	44.60
Lamb_2	67309706	99.81	97.28	94.02	93.66	97.78	44.36
Lamb_3	76906026	99.82	97.12	93.99	93.47	97.71	44.24
Adult_1	73783056	99.82	97.12	93.99	93.65	97.78	43.76
Adult_2	78182148	99.85	97.28	94.11	93.95	97.93	44.82
Adult_3	76149668	99.83	97.13	93.73	93.83	97.85	44.60

Note: In order to ensure data quality, the original data should be filtered before information analysis to reduce the analysis interference caused by invalid data. First of all, we use fastp to control the quality of raw reads from the machine, to filter low-quality data, and to obtain clean reads.

**Table 2 animals-13-00665-t002:** LncRNA-mRNA interaction analysis.

	Antisense			Cis		
	lncRNA	mRNA	Pair	lncRNA	mRNA	Pair
All	539	402	561	1973	1840	2625
Differently expressed	5	4	5	30	27	30

## Data Availability

The data presented in this study are available on request from the corresponding author. The data are not publicly available to preserve privacy of the data. All data relevant to the study has been submitted to the SRA database, study accession number: PRJNA891119. The data is accessible at the following link: https://www.ncbi.nlm.nih.gov/sra/PRJNA891119. Data are also available upon request from the authors.

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
