# Peer review of "Insights into Transcriptomic Differences in Ovaries between Lambs and Adult Sheep after Superovulation Treatment"

_animals, 2023, doi:10.3390/ani13040665_

Round 1
Reviewer 1 Report
Review - Manuscript ID animals-2114748
The objective of the manuscript “Insight of transcriptomic differences in ovary between lambs and adult sheep after superovulation treatment” was to study differentially expressed long non-coding RNA (LncRNAs) and messenger RNAs (mRNAs) in the ovaries of young and adult sheep. It´s an interesting study in which the authors could identify differentially expressed genes related to ovarian/follicle development and ovulation in lambs and adult sheep.
General comments
Simple summary
Lines 11-13 – The authors mentioned that the oocyte mutation ratio and the level of late embryonic development significantly decreased in the lamb superovulation groups; however these variables were not evaluated in this manuscript. Moreover, it is not clear if these results were found by the authors in another study. Thus, I suggest rewriting this part of the simple summary.
Lines 16-18 – Different acronyms refer to long non-coding RNA (lncRNA, lncRNA). This acronym must be standardized throughout the text.
Abstract
Line 32 – As pointed out in the discussion, the quality of mature follicles will be worse in lambs than that of adult sheep (lines 307-310). Thus, I think the conclusion “and leading to more lambs” is not adequate and should be deleted.
Introduction
Lines 71-72 – The authors affirm that the mechanism of superovulation in sheep is unclear. The authors should explain further and include literature supporting this statement, as well as should discuss the differences between lambs and adult sheep.
Line 77 – Change miRNAs by lncRNAs.
Materials and methods
The description on lines 94-105 refers to the methodology adopted in a previous study published in 2021 (Zhang, X., et al., The Roles of the miRNAome and Transcriptome in the Ovine Ovary Reveal Poor Efficiency in Juvenile 383 Superovulation. Animals (Basel), 2021. 11(1)). This study appears to be a continuation of the one already published; so the authors must make this clear.
Lines 96-97 – When was FSH applied (days of the superovulation protocol)?
For a better understanding of this part of the methodology, a timeline of the procedures (superovulation, blood samples, and ovaries recovery) could be presented in the manuscript.
Line 104 - The evaluation of the number of ovarian follicles was not described in the methodology. Were the follicles classified into different classes according to their diameter (eg < 3 mm, 3-5 mm, > 3 mm)? What was the number of follicles in each class in lambs and adult sheep? Could this information be useful in explaining differential gene expression in lambs and adult sheep?
Lines 107-109 - What was the sensitivity of the radioimmunoassay kits used for the determination of serum FSH, luteinizing hormone (LH), progesterone (P4), and estradiol (E2) concentrations? What was the intra-assay coefficient of variation for each hormone?
A statistical analysis description (item 2.9) must be included in the materials and methods section.
Results
The results presented in Table 1 were published in a previous paper (Zhang, X., et al., The Roles of the miRNAome and Transcriptome in the Ovine Ovary Reveal Poor Efficiency in Juvenile 383 Superovulation. Animals (Basel), 2021. 11(1)). I think that the same results should not be published again. Thus, I suggest removing item 3.1 from the results section. However, the results shown in Table 1 could be used in the discussion, with the citation of the paper in which they were published.
Table 2 is not self-explanatory. I suggest adding a description note of how each of the parameters shown in the table was calculated, as well as the meaning of the acronyms.
Throughout the text, write out the full name of the acronyms when first mentioned (with the acronym in brackets). Use only the acronym after this.
Line 199 – Change “lamps” to “lambs”
Figure 2 – In general, Figure 2C is hard to understand. What do the green and red markings on some DEmRNA represent? I suggest including a short description of the pathway shown and how it differs between lambs and adult sheep.
Figure 3 – Figure 3 shows a linear regression and its coefficient of determination (R2). This result needs to be described properly in the results section. Furthermore, this analysis must be described in the statistical analysis item.
Discussion
Lines 242-246 – The follicular and hormonal serum concentration data were published in a previous paper. As suggested, the authors should mention this in the discussion section.
Lines 248-251 – “...hormonal inducement of prepubescent donors needed smaller hormonal doses compared to adult animals, because of lower body weight and better follicular response prior to the emergence of intraovarian processes that control ovulation rate.” The authors should better explain why lambs have a better follicular response before the emergence of intraovarian processes that control ovulation rate.
Line 270 – LDLR?
Lines 269-271 - "... adult sheep had lower LDLR gene expression, higher LRP1 gene expression, and lambs LDL gene expression was higher, and LRP1 gene expression was lower.” The authors should better discuss these results. What is the physiological implication of these observations?
Line 274 – Delete the acronym GnRH.
Line 288 – I suggest changing “excrete” to “ovulate”.
Conclusion
The conclusions should be drawn based on the results obtained in this study. Thus, I suggest removing the first sentence from the conclusion.
Line 321 - I think the conclusion “and leading to more lambs” is not adequate. Thus, I suggest deleting it. According to lines 307-310, the quality of mature follicles will be worse in lambs than that in adult sheep. Thus, the higher number of ovulation would not necessarily result in more lambs.
Author Response
Point 1: Lines 11-13 – The authors mentioned that the oocyte mutation ratio and the level of late embryonic development significantly decreased in the lamb superovulation groups; However, these variables were not evaluated in this manuscript. Moreover, it is not clear if these results were found by the authors in another study. Thus, I suggest rewriting this part of the simple summary.
Response 1: Thank you for your good suggestions.Thanks.According to your suggestion, we rewrited this part, the part has been modified to “Nevertheless, as our previously published article reveals that oocytes derived from juvenile animals tend to be of poor quality, which means that they are less likely to mature and develop into embryos”
Point 2: Lines 16-18 – Different acronyms refer to long non-coding RNA (lncRNA, lncRNA). This acronym must be standardized throughout the text.
Response 2: Thanks.The format of acronyms in full text sentences has been modified LNCRNA to lncRNA as standardized required.
Point 3: Line 32 – As pointed out in the discussion, the quality of mature follicles will be worse in lambs than that of adult sheep (lines 307-310). Thus, I think the conclusion “and leading to more lambs” is not adequate and should be deleted.
Response 3: Thanks. We have deleted the conclusion “and leading to more lambs” .
Point 4: Lines 71-72 – The authors affirm that the mechanism of superovulation in sheep is unclear. The authors should explain further and include literature supporting this statement, as well as should discuss the differences between lambs and adult sheep.
Response 4: Thank you for your good suggestions. The mechanism of superovulation in sheep is not clear. We cite other literature to illustrate. During the past few years, a lot of research has been conducted on sheep breeding,but the mechanism of causing different physiological states of sheep's ovary after su-perovulation and estrus synchronisation treatment has not been fully understood.The development of oocytes and embryos from juvenile animals remains at a reduced level compared to that of adults, although more follicles can be acquired in juvenile animals.
Point 5: Line 77 – Change miRNAs by lncRNAs.
Response 5: Thanks. The miRNAs have been modifited to lncRNAs .
Point 6: The description on lines 94-105 refers to the methodology adopted in a previous study published in 2021 (Zhang, X., et al., The Roles of the miRNAome and Transcriptome in the Ovine Ovary Reveal Poor Efficiency in Juvenile 383 Superovulation. Animals (Basel), 2021. 11(1)). This study appears to be a continuation of the one already published; so the authors must make this clear.
Response 6: Thanks. We have revised the test method and explained the superovulation scheme in detail. This part has been modified to “The experimental grouping and superovulation treatment scheme in this paper was performed as described in our team previous published studies. We used the Chinese Hu sheep treated with superovulation , and two groups were set: the lamb group (1-month-old, n=6) and the adult group (24-month-old, n=6) ,serum and ovaries were collected.
Point 7: Lines 96-97 – When was FSH applied (days of the superovulation protocol)?
For a better understanding of this part of the methodology, a timeline of the procedures (superovulation, blood samples, and ovaries recovery) could be presented in the manuscript.
Response 7: Thanks. According to your suggestion, we added the superovulation scheme and the good use time and dose of FSH, “In order to minimize backgrounds caused by non-specific antibody binding, serum samples should generally be diluted at least 1:50.Serum product needs to be stored in a low temperature refrigerator of - 20 ℃ - 70 ℃ for a long time, other drugs should be stored in a refrigerator of 4 ℃, and the iodine [125I] radioimmunoassay kit should be stored in a refrigerator of - 20 ℃.”
Point 8: Line 104 - The evaluation of the number of ovarian follicles was not described in the methodology. Were the follicles classified into different classes according to their diameter (eg < 3 mm, 3-5 mm, > 3 mm)? What was the number of follicles in each class in lambs and adult sheep? Could this information be useful in explaining differential gene expression in lambs and adult sheep?
Response 8: Thank you for your good suggestions. The experimental time for the evaluation of ovarian follicle number was too early. There was no diameter statistics at that time. These data were published by our laboratory .
Point 9: Lines 107-109 - What was the sensitivity of the radioimmunoassay kits used for the determination of serum FSH, luteinizing hormone (LH), progesterone (P4), and estradiol (E2) concentrations? What was the intra-assay coefficient of variation for each hormone?
Response 9: Thanks. We use commercial hormone assay kits with reliable sensitivity and small error.
Point 10: A statistical analysis description (item 2.9) must be included in the materials and methods section.
Response 10: Thanks. We have added statistical analysis to the article. “An analysis of categorical variables, such as follicle numbers, was performed with the chi-square test. On SAS 8.0(SAS Institute Inc., Cary, NC, USA), Duncan's multiple range test program in ANOVA was used to analyze continuous variables of hormone concentration and mRNA/lncRNA expression. The results are presented as mean + standard error. As part of the analysis, the Spearman correlation was calculated based on SPSS version 20.0 (SPSS Inc., Chicago, IL, USA) to assess the relationship between high-throughput sequencing and real-time PCR in mRNA/miRNA expression assays.”
Point 11: The results presented in Table 1 were published in a previous paper (Zhang, X., et al., The Roles of the miRNAome and Transcriptome in the Ovine Ovary Reveal Poor Efficiency in Juvenile 383 Superovulation. Animals (Basel), 2021. 11(1)). I think that the same results should not be published again. Thus, I suggest removing item 3.1 from the results section. However, the results shown in Table 1 could be used in the discussion, with the citation of the paper in which they were published.
Response 11: Thanks. We deleted item 3.1 from the results section. The results shown in Table 1 was used in the discussion, with the citation of the paper in which they were published.
Point 12: Table 2 is not self-explanatory. I suggest adding a description note of how each of the parameters shown in the table was calculated, as well as the meaning of the acronyms.
Response 12: Thanks. We have added a description below Table 2“Note: In order to ensure data quality, the original data should be filtered before information analysis to reduce the analysis interference caused by invalid data. First of all, we use fastp to control the quality of raw reads off the machine, filter low-quality data, and get clean reads.”
Point 13: Throughout the text, write out the full name of the acronyms when first mentioned (with the acronym in brackets). Use only the acronym after this.
Line 199 – Change “lamps” to “lambs”
Response 13: Thanks.We have modifited the acronym error and Change “lamps” to “lambs” .
Point 14: Figure 2 – In general, Figure 2C is hard to understand. What do the green and red markings on some DEmRNA represent? I suggest including a short description of the pathway shown and how it differs between lambs and adult sheep.
Response 14:. Thanks.The heatmap of DE mRNAs involved in the ovarian steroidogenesis pathway; the greater the expression level in red color, and the lesser the expression level in bluer the color.
Point 15: Figure 3 – Figure 3 shows a linear regression and its coefficient of determination (R2). This result needs to be described properly in the results section. Furthermore, this analysis must be described in the statistical analysis item.
Response 15: Thanks. We have added the results descirption” The RNA expression level was determined by the ∆∆‐2CT method. qRT‐PCR was repeated three times for each sample. GAPDH was used as a reference gene. The expression levels of LncRNAs and their target genes are depicted in Figure 2. The qRT‐PCR results used to validated the RNA‐seq results. ”
Point 16: Lines 242-246 – The follicular and hormonal serum concentration data were published in a previous paper. As suggested, the authors should mention this in the discussion section.
Response 16: Thanks.The follicular and hormonal serum concentration data have been mentioned in the discussion section.
Point 17: Lines 248-251 – “...hormonal inducement of prepubescent donors needed smaller hormonal doses compared to adult animals, because of lower body weight and better follicular response prior to the emergence of intraovarian processes that control ovulation rate.” The authors should better explain why lambs have a better follicular response before the emergence of intraovarian processes that control ovulation rate.
Response 17: Thank you for your good suggestions. In previous study indicate that FSHR and LHCGR expression levels in lamb ovaries were significantly higher than those in adult sheep ovaries. Specifically, at the same hormone level, lambs have an increased sensitivity to hormones due to the increased expression of hormone receptors in ovarian tissue. Based on RT-PCR, immunohistochemistry, and Western blot results, Scarlet et al. also showed that FSHR was present in the ovaries of prepubertal fillies, and found that FSHR was abundant in the ovarian stroma cells of neonate fillies but not of adults. (Scarlet D., Walter I., Hlavaty J., Aurich C. Expression and immunolocalisation of follicle-stimulating hormone receptors in gonads of newborn and adult female horses. Reprod. Fertil. Dev. 2016;28:1340–1348. doi: 10.1071/RD14392.)
Point 18: Line 270 – LDLR?
Response 18: Thanks.The LDLR is the LDL receptor abbreviation.
Point 19: Lines 269-271 - "... adult sheep had lower LDLR gene expression, higher LRP1 gene expression, and lambs LDL gene expression was higher, and LRP1 gene expression was lower.” The authors should better discuss these results. What is the physiological implication of these observations?
Response 19: Thanks. We have take a disscusion about the physiological effects of LDLR in preceding text .Across receptor-mediated endocytosis of lipoprotein particles, the LDL receptor regulates cholesterol homeostasis. Enhanced expression of LRP deliver cholesterol for steroidogenesis, and it represent a backup system to guarantee the pathway transfers cholesterol to steroidogenic tissues. For steroidogenesis, rat ovarian granulosa depends largely on lipoprotein-derived cholesterol, primarily provided by the LDL receptor- and scavenger receptor class B type I (SRBI)-mediated pathways.This may be the mechanism of the influence of hormone regulation on physiological effects
Point 20: Line 274 – Delete the acronym GnRH.
Response20: Thanks. The acronym GnRH has been deleted.
Point 21: Line 288 – I suggest changing “excrete” to “ovulate”.
Response 21: Thanks. We have changed “excrete” to “ovulate”.
Point 22: The conclusions should be drawn based on the results obtained in this study. Thus, I suggest removing the first sentence from the conclusion.
Response 22: Thanks. We have deleted this sentence.
要点23:321 行 - 我认为“并导致更多的羔羊”的结论是不够的。因此,我建议删除它。根据307-310行,羔羊的成熟卵泡质量将比成年绵羊差。因此,较高的排卵次数不一定会导致更多的羔羊。
回应23:感谢您的好建议。根据您的建议,我们删除了“并导致更多的羔羊”。

Reviewer 2 Report
I have read and reviewed your manuscript entitled "Insight of transcriptomic differences in ovary between lambs and adult sheep after superovulation treatment ". In my opinion, the Authors present valuable data, however, the present form of the paper does not fit for publication in the Animals journal. The list of my comments and suggestions is presented below:
1. Please add a section explaining all the statistical analyses you used in your studies.
2. Chapter 2.3. was specified enough, please provided information such as: how the serum was stored, serum dilution, etc.
3. Chapter 2.8. This chapter does not follow the MIQE guidelines for publication of qPCR results (primers sequence and concentration, method thermal profile and so on). Have you performed primers validation?
4. Figure 2 is illegible
5. Whole manuscript is a bit chaotic and messy. There is a lot of editorial errors, i.e. “E2”/”E2”, vs. 2.0 bold / without bold, gene names with italic/non-italic, etc.
Author Response
Point 1: Please add a section explaining all the statistical analyses you used in your studies.
Response 1: Thank you for your good suggestions. According to your suggestion, We have added statistical analysis to the article. “An analysis of categorical variables, such as follicle numbers, was performed with the chi-square test. On SAS 8.0(SAS Institute Inc., Cary, NC, USA), Duncan's multiple range test program in ANOVA was used to analyze continuous variables of hormone concentration and mRNA/lncRNA expression. The results are presented as mean + standard error. As part of the analysis, the Spearman correlation was calculated based on SPSS version 20.0 (SPSS Inc., Chicago, IL, USA) to assess the relationship between high-throughput sequencing and real-time PCR in mRNA/miRNA expression assays.”
Point 2: Chapter 2.3. was specified enough, please provided information such as: how the serum was stored, serum dilution, etc.
Response 2: Thank you for your good suggestions. According to your suggestion, methods such as how to store serum and serum dilution have been added to the materials and methods.
Point 3: Chapter 2.8. This chapter does not follow the MIQE guidelines for publication of qPCR results (primers sequence and concentration, method thermal profile and so on). Have you performed primers validation?
Response 3: Thanks.We have performed primers validation in previous study, therefore, it is not discussed in this paper
Point 4: Figure 2 is illegible.
Response 4: Thank you for your good suggestions. According to your suggestion, we have adjusted the resolution of the Figure .
Point 5: Whole manuscript is a bit chaotic and messy. There is a lot of editorial errors, i.e. “E2”/”E2”, vs. 2.0 bold / without bold, gene names with italic/non-italic, etc.
Response 5: Thank you for your good suggestions. According to your suggestion, We have changed the corner mark of the article, and italic/non italic, bold/non bold, etc.

Reviewer 3 Report
Introduction
The introduction section is premature and not containing all references that are related to the study. The author should expand the introduction section and cite all studies that are related to their study
Materials and methods
Generally, the authors did not explain the methods that have used in their study. Since not all authors are familiar with the used methods, they should be explained in the manuscript.
What was the RNA integrity number for the samples used in the study?
Discussion,
Generally, the discussion is weak, and the authors did not discuss their results enough.
The author claims that after superovulation, there was no non-significant variation in the serum levels of FSH and LH between the two groups, where lamb’s ovulation was significantly higher than adult sheep. However, the authors did not observe any difference in FSH, LH, P4, and E2 serum levels in both groups. It is interesting that lambs have low levels of estrogen and progesterone in their blood despite having a high number of follicles in their ovaries. What could be the plausible reason for this? Authors should discuss it.
The conclusion is not clear. Authors should write a descriptive and all-encompassing conclusion that covers all their results.
Author Response
Point 1: Introduction
The introduction section is premature and not containing all references that are related to the study. The author should expand the introduction section and cite all studies that are related to their study
Response 1: Thank you for your good suggestions. According to your suggestion, we have added related contents “During the past few years, a lot of research has been conducted on sheep breeding[17], but the mechanism of causing different physiological states of sheep's ovary after superovulation and estrus synchronisation treatment has not been fully understood[18]. The development of oocytes and embryos from juvenile animals remains at a reduced level compared to that of adults, although more follicles can be acquired in juvenile animals.”
Point 2: Generally, the authors did not explain the methods that have used in their study. Since not all authors are familiar with the used methods, they should be explained in the manuscript.
Response 2: Thank you for your good suggestions. According to your suggestion,We has revised the test method and explained the superovulation scheme in detail. This part has been modified to “The experimental grouping and superovulation treatment scheme in this paper was performed as described in our team previous published studies[19]. We used the Chinese Hu sheep treated with superovulation , and two groups were set: the lamb group (1-month-old, n=6) and the adult group (24-month-old, n=6) ,serum and ovaries were collected.
Point 3: What was the RNA integrity number for the samples used in the study?
Response 3: Thanks. We has confirmed that samples RNA integrity quality control RIN value are not less than 7 before sequencing.
Point 4: Generally, the discussion is weak, and the authors did not discuss their results enough.
Response 4: Thank you for your good suggestions. According to your suggestion,We has added the discussion on the corresponding results section.“Compared with adult sheep, gonadotropin superovulation to lambs can stimulate follicle growth and obtain more oocytes for in vitro embryo production[Armstrong et.al, Reprod Fertil Dev. 1997 ], which has a great application in animal husbandry. However, in vitro embryonic development potential derived from lamb oocytes is significantly reduced [Readeret.al, Reprod Fertil Dev. 2015; Gou et.al, Anim Reprod Sci. 2009;112:316–324. doi: 10.1016/j.anireprosci.2008.05.008.]. The causes of oocyte dysplasia in juvenile animals are not fully understood and remain to be studied. In this study, we identified lncRNA sets and transcriptome data from the ovaries of overostracized lambs and adult sheep to explore the mechanism of lncRNA development in lamb oocytes.We has screened 242 DE lncRNAs and 3150 DE mRNAs in the ovaries of lamb and adult sheep.Through GO and KEGG analysis, we correlated these DE mRNAs related to ovarian/follicle development and ovulation , including OaFSHR; OaLHCGR; OaLDLR; OaZP3; OaSCARB1; OaPDGFRA; while through lncRNA-mRNA correlation analysis, we found XR_003585520.1, MSTRG.15652.1, XR_003588840.1, and their paired genes; PDGFC, LRP5, LRP1were assoscited to ovarian/follicle development and ovulation .We also observed a synergistic effect between PDGFR and LRP1. Among the three lncRNAs, we found that XR_003588840 that significantly different and might perform a regulatory part in ovarian/follicle growth or ovulation.”
Point 5: The author claims that after superovulation, there was no non-significant variation in the serum levels of FSH and LH between the two groups, where lamb’s ovulation was significantly higher than adult sheep. However, the authors did not observe any difference in FSH, LH, P4, and E2 serum levels in both groups. It is interesting that lambs have low levels of estrogen and progesterone in their blood despite having a high number of follicles in their ovaries. What could be the plausible reason for this? Authors should discuss it.
Response 5: Thank you for your good suggestions. In our previous study indicate that FSHR and LHCGR expression levels in lamb ovaries were significantly higher than those in adult sheep ovaries. Specifically, at the same hormone level, lambs have an increased sensitivity to hormones due to the increased expression of hormone receptors in ovarian tissue.
Point 6: The conclusion is not clear. Authors should write a descriptive and all-encompassing conclusion that covers all their results.
Response 6: Thank you for your good suggestions. According to your suggestion,We has modified the conclusion to “This study use RNA‐seq data from superovulation treatment lambs and adult Hu sheep to classify LncRNAs and LncRNA target genes. According to sequencing data, OaPDGFR played a leading role when compared with OaLRP1, however, the synergistic effect between PDGFR and LRP1. Furthermore, the induced OaLHCGR in lambs is higher than that in adult sheep, so it may be more sensitive to LH and stimulate oocyte maturation . The LncRNAs may play an uniquely role in sheep prolificacy. Finally, we found that XR_003588840.1 was differentially expressed, indicating its potential role in ovarian/follicle development regulation. Our results identifying a possible candidate LncRNAs as well as an imortant role in hormones response, oocyte mature to address sheep prolificacy barriers.”

Round 2
Reviewer 1 Report
The authors were able to considerably improve the manuscript. However, I still suggest a few more changes outlined below.
Lines 10-12 – I suggest a small change in the text to “Nevertheless, our previous study revealed that oocytes derived from juvenile animals tend to be of poor quality, which means that they are less likely to mature and develop into embryos”.
Line 99, item 2.3 – As requested, results on serum hormone concentrations were removed from this manuscript because these results have already been previously published. However, the authors maintained the description of the hormonal dosage. Therefore, I suggest removing this item. Alternatively, the description of the measurement of the serum hormone concentrations could be, in short, mentioned in item 2.2.
Lines 159-161 – Statistical analysis item is not adequately described.
As requested, results on follicular development and serum hormone concentrations were removed from this manuscript because these results have already been previously published. However, the authors maintained the description of the analysis corresponding to these results in the item statistical analysis. Therefore, I suggest rewriting this item by removing the description of the analyzes whose results are not presented in this manuscript.
On the other hand, Pearson’s correlation analysis was mentioned in line 175 but was not described in the statistical analysis item.
Please, review the following text: “As part of the analysis, the Spearman correlation was calculated based on SPSS version 20.0 (SPSS Inc., Chicago, IL, USA) to assess the relationship between high-throughput sequencing and real-time PCR in mRNA/miRNA expression assays.” It looks like there was a mistake in the description of the analysis.
Author Response
Dear Editor:
Thank you very much for giving us comments on our manuscript, entitled“Insight of transcriptomic differences in ovary between lambs and adult sheep after superovulation treatment”(animals-2114748). The comments and suggestions from the Reviewers were valuable and helpful for improving our manuscript. Here, the comments and suggestions from the Reviewers are all addressed point by point as follows. We have highlighted all the changes in our manuscript and have resubmitted our revised manuscript to Animals.
Response 1: Thanks. I have changed in the text to “Nevertheless, our previous study revealed that oocytes derived from juvenile animals tend to be of poor quality, which means that they are less likely to mature and develop into embryos”.
Point 2: Line 99, item 2.3 – As requested, results on serum hormone concentrations were removed from this manuscript because these results have already been previously published. However, the authors maintained the description of the hormonal dosage. Therefore, I suggest removing this item. Alternatively, the description of the measurement of the serum hormone concentrations could be, in short, mentioned in item 2.2.
Response 2: Thank you for your good suggestions. According to your suggestion, we have delete" item 2.3, Serum hormone determination". The description of serum hormone concentration measurement " Serum product was utilized to determine the levels of FSH, luteinizing hormone (LH), progesterone (P4), and estradiol (E2). Each hormone was tested employing iodine [125I] radioimmunoassay kit (BNIBT, Beijing, China)" is added in 2.2
Point 3: Lines 159-161 – Statistical analysis item is not adequately described.
As requested, results on follicular development and serum hormone concentrations were removed from this manuscript because these results have already been previously published. However, the authors maintained the description of the analysis corresponding to these results in the item statistical analysis. Therefore, I suggest rewriting this item by removing the description of the analyzes whose results are not presented in this manuscript.
Response 3: Thanks. We have rewritten and fully described the statistical analysis item. " In order to verify the reliability of transcriptome data, we analyzed the correlation between RNA-seq data and qRT-PCR data. In this study, each sample was tested three times.The qRT-PCR data used ΔΔCt (ΔCt reference-ΔCt target) and 2-ΔΔCt formulas to obtain standardized gene expression .The relevant fitting curve data between qRT-PCR and RNA-seq gene expression results are from supplementary file 1.Finally, SPSS (v 20.0) was used to statistically analyze the experimental data".
Point 4: On the other hand, Pearson’s correlation analysis was mentioned in line 175 but was not described in the statistical analysis item.
Response 4: Thanks. Pearson correlation analysis in line 175, but it was not described in the statistical analysis items,so we have deleted this section.
要点5:请查看以下文本:“作为分析的一部分,Spearman相关性基于SPSS版本20.0(SPSS Inc.,芝加哥,伊利诺伊州,美国)计算,以评估mRNA / miRNA表达测定中的高通量测序与实时PCR之间的关系。看起来分析的描述有误。
回应 5:谢谢。因为这部分的描述是错误的,我们改写为“ 为了验证转录组数据的可靠性,我们分析了RNA-seq数据与qRT-PCR数据之间的相关性。在这项研究中,每个样品都进行了三次测试。qRT-PCR数据使用ΔΔCt(ΔCt参考-ΔCt靶标)和2-ΔΔCt公式获得标准化基因表达。qRT-PCR与RNA-seq基因表达结果的相关拟合曲线数据来自补充文件1,最后采用SPSS(v 20.0)对实验数据进行统计分析。

Reviewer 2 Report
I have read and reviewed your manuscript entitled "Insight of transcriptomic differences in ovary between lambs and adult sheep after superovulation treatment ". Thank you for making all corrections according to my comments and I accept this manuscript in its present form.
Author Response
Thank you for your affirmation, and at the same time, thank you for your careful review,your valuable comments have benefited me a lot. Thank you very much.I wish you success in your work.